# The Soybean *bZIP* Transcription Factor Gene *GmbZIP2* Confers Drought and Salt Resistances in Transgenic Plants

**DOI:** 10.3390/ijms21020670

**Published:** 2020-01-20

**Authors:** Yan Yang, Tai-Fei Yu, Jian Ma, Jun Chen, Yong-Bin Zhou, Ming Chen, You-Zhi Ma, Wen-Liang Wei, Zhao-Shi Xu

**Affiliations:** 1College of Agriculture, Yangtze University, Hubei Collaborative Innovation Center for Grain Industry, Engineering Research Center of Ecology and Agricultural Use of Wetland, Ministry of Education, Jingzhou 434025, China; 13545687096@163.com; 2Institute of Crop Science, Chinese Academy of Agricultural Sciences (CAAS), National Key Facility for Crop Gene Resources and Genetic Improvement, Key Laboratory of Biology and Genetic Improvement of Triticeae Crops, Ministry of Agriculture, Beijing 100081, China; yutaifei824@163.com (T.-F.Y.); chenjun@caas.cn (J.C.); zhouyongbin@caas.cn (Y.-B.Z.); chenming02@caas.cn (M.C.); mayouzhi@caas.cn (Y.-Z.M.); 3College of Agronomy, Jilin Agricultural University, Changchun 130118, China; majian197916@jlau.edu.cn

**Keywords:** bZIP transcription factor, expression pattern, gene regulate, abiotic stress resistance, soybean

## Abstract

Abiotic stresses, such as drought and salt, are major environmental stresses, affecting plant growth and crop productivity. Plant bZIP transcription factors (bZIPs) confer stress resistances in harsh environments and play important roles in each phase of plant growth processes. In this research, 15 soybean bZIP family members were identified from drought-induced de novo transcriptomic sequences of soybean, which were unevenly distributed across 12 soybean chromosomes. Promoter analysis showed that these 15 genes were rich in ABRE, MYB and MYC *cis*-acting elements which were reported to be involved in abiotic stress responses. Quantitative real-time polymerase chain reaction (qRT-PCR) analysis indicated that 15 *GmbZIP* genes could be induced by drought and salt stress. *GmbZIP2* was significantly upregulated under stress conditions and thus was selected for further study. Subcellular localization analysis revealed that the GmbZIP2 protein was located in the cell nucleus. qRT-PCR results show that *GmbZIP2* can be induced by multiple stresses. The overexpression of *GmbZIP2* in *Arabidopsis* and soybean hairy roots could improve plant resistance to drought and salt stresses. The result of differential expression gene analysis shows that the overexpression of *GmbZIP2* in soybean hairy roots could enhance the expression of the stress responsive genes *GmMYB48*, *GmWD40*, *GmDHN15*, *GmGST1* and *GmLEA*. These results indicate that soybean *bZIP*s played pivotal roles in plant resistance to abiotic stresses.

## 1. Introduction

When plants suffer abiotic stresses such as drought and salt, the growth processes of plants are often seriously affected. Adverse environments can dramatically reduce crop yields [1,2,3,4]; therefore, it is extremely urgent to increase plant resistances to abiotic stresses. A series of sophisticated strategies for survival have been developed in plants which ensure plant acclimation to adverse environments [5,6,7,8]. Changes in some functional genes at the transcriptomic level are conducive to improving plants’ resistance. These genes, such as transcription factors, protein kinases and protein phosphatases, are responsible for transducing stress signals and regulating the expression of stress-responsive genes that can produce or regulate the enzymes that are involved in the biosynthesis of various osmoprotectants and subsequently late-embryogenesis abundant glutathione S-transferases, proteins that counteract environmental damage [9,10,11]. Among these genes, transcription factors, as a part of the ultimate regulator, bind to gene promoters to trigger downstream gene transcription [12].

Plant transcription factors, as primary regulators, induce gene expression and modify the temporal and spatial expression patterns of responsive genes, which play an important part in plant growth cycle [13,14,15]. Transcription factors are represented in massive numbers and in many different groups in plant genomes and they are identified and classified according to their DNA-binding domains (DBDs). Therefore, the names of the DBDs are used to define transcription factor families [16] and their crucial functions in plant processes have been verified to resist abiotic stresses [16,17]; in this way, plant transcription factors were identified and classified into many families, such as DREB, WRKY, bZIP, NF-Y and NAC [18,19]. Past research has demonstrated that these plant transcription factor families act as a crucial part in plant resistances to abiotic stresses. For example, plant DREB transcription factors bind to a dehydration responsive element (DRE) *cis*-element in the promoter region of several stress-responsive genes in response to dehydration, high salinity or low temperature [20,21] and research has also proved that the overexpression of DREB transcription factors in plants could improve transgenic plant resistance to abiotic stresses [20]. Similarly, WRKY transcription factors containing WRKY domains have high binding affinity to the consensus *cis*-acting element termed the W box (TTGACT/C), which regulates the expression of stress-responsive genes [22].

Plant bZIPs contain a highly conserved region composed of a leucine zipper and adjacent basic amino acids. This leucine zipper, containing a periodic repetition of leucine residues at every seventh position, is a helical domain. Two leucine zipper domains form a parallel, helical-coiled coil, which is stabilized by hydrophobic interactions. This dimerization juxtaposes two basic regions to form the DNA binding site [23,24]. Many *bZIP* transcription factor families have been identified in different plant species, such as maize, cucumber and leguminous plants [25,26,27]. Plant bZIPs are unveiled be essential for diverse biological processes in plants, such as seed maturation, flower development and stress signaling transduction [28]. A recent discovery indicated that the accumulation of *OsbZIP73* facilitates the adaptation of japonica rice to cold climates [29]. *Arabidopsis AtbZIP17* and *AtbZIP28* regulate root elongation during stress response [30].

Soybean (*Glycine max*) is a dicotyledonous plant which is widely cultivated in Northern China, the United States, Brazil and Argentina. It serves as a vital food and oil crop in arid regions. Despite previous studies identifying 138 family members of soybean *bZIP* transcription factors [27], their functions during plant resistance to abiotic stresses still remain largely elusive. In this work, we analyzed drought-induced de novo transcriptomic sequences of soybean and found 15 upregulated drought-responsive *bZIP* family members in soybean. It was revealed by qRT-PCR analysis that *GmbZIP2* had higher transcriptional levels than the other *GmbZIP* genes after drought and salt treatments and this was thus chosen for further analysis. Subsequent analysis found that *GmbZIP2* responded to numerous stresses and could be induced by drought, salt, abscisic acid (ABA) and cold and the overexpression of *GmbZIP2* in plants improved their tolerance to drought and salt stresses.

## 2. Results

### 2.1. De novo Transcriptomic Sequences Analyses of Soybean

To elucidate the function of bZIPs under stress conditions, four-leaf stage soybean seedlings underwent drought treatment for 2 h and then were used for de novo transcriptomic sequence analyses. The data analysis from the de novo transcriptome sequencing showed that the transcriptional levels of some genes were changed before or after the drought treatment. According to the functional annotation of differentially expressed genes, we found that 15 members of the soybean *bZIP* genes were induced to be upregulated (Appendix A) and then were selected for further research.

### 2.2. Sequence Analysis of GmbZIPs in Soybean

The 15 soybean bZIPs were identified by sequence alignment from the completed soybean genome sequence. Previous research showed that 138 soybean *bZIP* transcription factor family members (actually updated to 136) were identified and were divided into 12 groups [27]. Here, the 15 upregulated soybean *bZIP*s were distributed among six groups (Appendix A). The established evolutionary relationships indicated that *GmbZIP1*, *GmbZIP4*, *GmbZIP5*, *GmbZIP6*, *GmbZIP7* and *GmbZIP12* were closer than the others (evolutionary branches were shorter than the others) (Figure 1 and Appendix A). Out of these 15 soybean *bZIP*s, 13 members contained two to eleven introns and only *GmbZIP2* and *GmbZIP3* did not have any intron (Figure 2A). The structures of these 15 GmbZIP proteins contained the basic region leucin zipper (BRLZ) domain (Figure 2B and Appendix A). Most of the 15 GmbZIP proteins contained low-complexity region (LCR) domains and only GmbZIP2, GmbZIP3 and GmbZIP14 proteins did not have any LCR domains (Figure 2B). These 15 *GmbZIP* genes were distributed across 12 chromosomes (Figure 3). Nine chromosomes of soybean contained one of the 15 *GmbZIP* genes, while chromosome 2, 8 and 16 each contained two of the 15 *GmbZIP* genes (Figure 3).

### 2.3. Cis-Acting Element Analysis of 15 Soybean bZIP Gene Promoters

*Cis*-element promoter analysis showed that all of the 15 soybean *bZIP* gene promoters contained MYB and MYC elements and most of the *GmbZIP* genes had ABA-responsive element (ABRE) *cis*-acting elements and only *GmbZIP8*, *GmbZIP13* and *GmbZIP15* promoters did not have ABRE *cis*-acting elements (Appendix A). In addition, among the 15 *GmbZIP* genes, only *GmbZIP2*, *GmbZIP8*, *GmbZIP10* and *GmbZIP14* contained the dehydration response element (DRE) *cis*-acting elements. In addition, 60% of the 15 soybean *bZIP* members contain low-temperature responsive element (LTR), 45% contain CGTCA-motif *cis*-acting element and 67% contain TGACG-motif *cis*-acting element (Appendix A). These results indicate that most soybean *bZIP* genes might play important roles in the abiotic stress responses.

### 2.4. Tissue-Specific Expression Patterns of 15 GmbZIPs

The expression pattern analysis of 15 soybean *bZIP* genes in different soybean tissues was analyzed. Our results reveal that *GmbZIP3*, *GmbZIP6*, *GmbZIP11 and GmbZIP15* had the highest expressions in various soybean tissues compared to the other *GmbZIP* members; in particular, *GmbZIP3* was highly expressed under nodule symbiotic conditions, high ammonia and high nitrate conditions in roots compared to other tissues in soybean (Figure 4). On the contrary, *GmbZIP2* and *GmbZIP13* had the lowest expression in various soybean tissues compared to the other 13 soybean *bZIP* transcription factors. However, high expression levels of *GmbZIP1*, *GmbZIP4*, *GmbZIP5 and GmbZIP10* were found in the flower, leaves, nodules, pod, root, root hairs, seed, stem and shoot apical meristem (Figure 4).

### 2.5. Expression Pattern Analysis of 15 Soybean bZIPs under Drought and Salt Stresses

To analyze the transcript levels of the 15 soybean *bZIP* transcription factors under different conditions, qRT-PCR was carried out by using RNA isolated from four-leaf stage seedings of stress-treated soybean. Our data showed that 14 of the *GmbZIP* genes were upregulated under drought stress, the only exception being *GmbZIP1*. Among these 14 soybean *bZIP* transcription factors, *GmbZIP2*, *GmbZIP8*, *GmbZIP12 and GmbZIP15* were upregulated more than 10-fold during 12 h of drought stress and the expression levels of *GmbZIP2* and *GmbZIP8* were significantly up-regulated 30-fold under drought treatment (Figure 5). Compared with the expression levels of the *GmbZIP3*, *GmbZIP4*, *GmbZIP5*, *GmbZIP6*, *GmbZIP7*, *GmbZIP9, GmbZIP10*, *GmbZIP11 and GmbZIP14* genes at 0 h, their expression levels under drought treatment were up-regulated approximately four-fold (Figure 5). Under the NaCl condition, our qRT-PCR results reveal that *GmbZIP3* and *GmbZIP15* were slightly up-regulated and *GmbZIP1*, *GmbZIP2*, *GmbZIP4*, *GmbZIP5*, *GmbZIP6 and GmbZIP7* were clearly induced and upregulated more than 15-fold in response to salt stress, especially *GmbZIP2* (Figure 6). However, the expression levels of *GmbZIP8*, *GmbZIP9*, *GmbZIP10*, *GmbZIP11*, *GmbZIP12*, *GmbZIP13 and GmbZIP14* were nearly unchanged (Figure 6). By analyzing the expression pattern of *GmbZIP*s under drought and salt stresses, we found that *GmbZIP2* had the highest transcript levels at 6 and 12 h of drought treatment and was induced by salt (Figure 5 and Figure 6), which indicates that *GmbZIP2* is responsive to drought and salt stresses. Thus, we focused our further research on *GmbZIP2*.

### 2.6. Molecular Characteristics of GmbZIP2 Gene

To explore the subcellular location of the soybean *GmbZIP2* gene, the 16,318-hGFP vector was used. The open reading frame (ORF) sequence without the stop codon of soybean *GmbZIP2* was fused to an hGFP reporter protein that was located at the N-terminus and was then cloned into the recombinant vector GmbZIP2-16318-hGFP and then transferred into *Arabidopsis* protoplast cells to observe the localization of GFP fluorescence. The subcellular localization of the encoded GmbZIP2 protein was determined and assessed by the transient expression assays in *Arabidopsis* protoplasts using translational fusions to GFP. The control GFP was localized to the plasma membrane, nucleus and cytosol, whereas GmbZIP2 localized to the nucleus only (Figure 7A). To further verify this result, the plasma membrane, cytosol and nucleus proteins of *Arabidopsis* protoplasts that contained the GFP signal were extracted and detected by the GFP antibody. The results again show that GmbZIP2 is localized to the cell nucleus (Figure 7B).

To analyze the *GmbZIP2* expression at the transcript level under stress conditions, qRT-PCR was carried out by using the RNA isolated from stress-treated soybean. *GmbZIP2* was induced not only by drought and salt but also by ABA, mannitol and cold (Figure 7C). In response to ABA, mannitol and cold, *GmbZIP2* peaked at 4 h after treatment and then dropped off over time. At 4 h of ABA, mannitol and cold stress, the *GmbZIP2* transcript levels were upregulated to 3.5 times, 6.15 times and 6.38 times the unstressed level, respectively. However, the transcription level of *GmbZIP2* showed no significant changes under heat treatment (Figure 7C). The results imply that *GmbZIP2* is responsive to multiple stresses.

### 2.7. Overexpression of GmbZIP2 in Arabidopsis Improved Drought and Salt Tolerances

To evaluate the tolerance of transgenic *Arabidopsi*s plants of drought and salt stresses, the control of the CaMV 35S promoter, three T3 homozygous lines of transgenic *GmbZIP2 Arabidopsis* were used for stress tolerance analysis. The transcription levels of *GmbZIP2* in these three transgenic *Arabidopsis* lines were detected by semi-quantitative PCR and qRT-PCR and the results show that *GmbZIP2* had elevated expression in all three transgenic lines (Figure 8A–C). As shown in Figure 8, the 3-week-old plants of three transgenic *Arabidopsis* lines and WT (wild-type, WT, Col-0) plants grow well and similarly under normal conditions (22 °C, light 16 h/dark 8 h, 50% of relative humidity). However, when the transgenic *Arabidopsis* lines and WT plants were exposed to drought, significant differences appeared and the transgenic lines grew remarkably well compared to the WT plants (Figure 8A). After two weeks of drought and salt treatments, approximately 80% of transgenic *Arabidopsis* plants survived, but, regarding WT plants, only approximately 10–20% survived (Figure 8D). The results of the physiological and biochemical indexes show that the transgenic *Arabidopsis* plants had a higher chlorophyll content and a lower malondialdehyde (MDA) content and relative electrical conductivity (REC) than WT plants (Figure 8E,F).

### 2.8. GmbZIP2 Improves Stress Tolerance in Transgenic Soybean Hairy Roots

To further verify the relationship between *GmbZIP2* and stress response, we generated transgenic soybean hairy root composite plants and found that the overexpression of *GmbZIP2* conferred enhanced resistance to drought and salt in the transgenic plants (Figure 9A,B). *GmbZIP2* was detected as being overexpressed in transgenic soybean hairy roots by qRT-PCR (Figure 9C,D). Root elongation was measured under control and stress conditions and the results show that the transgenic hairy root composite plants under stress conditions had significantly longer roots than the controls under drought and salt stress (Figure 9E–G). The physiological and biochemical indexes of transgenic *GmbZIP2* soybean hairy roots were detected and the results reveal that the transgenic *GmbZIP2* soybean hairy roots had a higher proline content and a lower MDA content and REC than the control plants under drought and salt stresses (Figure 9H–J).

### 2.9. Regulatory Mechanism Analysis of GmbZIP2 in Soybean

To investigate the possible molecular mechanisms of *GmbZIP2* during stress response, transgenic soybean hairy root lines and EV-control (the empty plasmid of pCAMBIA3301) plants were used for analyzing the differential expression of five stress-responsive genes, namely *GmMYB48*, *GmWD40*, *GmDHN15*, *GmGST1 and GmLEA*, which were reported to act either directly or indirectly in abiotic stress responses and from our *de novo* transcriptomic sequences analyses of soybean (Appendix A). Analysis of cis-acting elements showed that this five stress-responsive gene promoters contained ACGT elements that could be bound by bZIP transcription factors (Appendix A). qRT-PCR assays of five stress response-related genes were performed after the transgenic soybean hairy root lines and EV-control plants were treated with 150 mM NaCl and 200 mM mannitol, using corresponding untreated lines (25 °C, light 16 h/dark 8 h, 50% of relative humidity) as controls. A two-fold change in expression was arbitrarily considered an induction of expression. qRT-PCR data showed that the expression levels of these five stress-responsive genes were upregulated in the transgenic *GmbZIP2* soybean hairy root plants compared with the EV-control plants under normal growth conditions and moreover, the expression levels of these five stress-responsive genes were dramatically upregulated under drought treatment compared with the control plants (Figure 10).

## 3. Discussion

Basic region leucine zippers are conserved in eukaryon over the course of their evolution and exist in plants, where they are associated with various functions. They exist in high numbers in different species—53 bZIP family members have been found in human beings [31], 136 in soybean [27], 78 in *Arabidopsis* [32] and 247 in rapeseed [33]. In previous works, bZIP family members were classified into 12 groups in *Arabidopsis* and soybean according to their conserved DNA-binding domains (DBDs) [27,32,34]. However, here we re-divided the bZIP family members of *Arabidopsis* and soybean into six groups according to their conserved amino acid sequences. We found 15 soybean bZIP family members were detected by de novo transcriptome sequencing and nine GmbZIP members were clustered in group I, two in group II, three in group III and only one in group IV, respectively. *GmbZIP1*, *GmbZIP2*, *GmbZIP3*, *GmbZIP4*, *GmbZIP5*, *GmbZIP6*, *GmbZIP7*, *GmbZIP12*
*and GmbZIP15* were classified into group I and had the highest level of homology with the *Arabidopsis* GBF (G-box factors) family of bZIP proteins [32]. GBF-type bZIP transcription factors were reported to associate with plant responses to hormones such as abscisic acid (ABA), ethylene and methyl jasmonate (MeJA), which play important roles in plant resistances [35]. *GmbZIP8* and *GmbZIP14* was classified into the group II with *AtbZIP39*, which were ABI5 proteins and responded to ABA. In group II, *AtbZIP35*, *AtbZIP37* and *AtbZIP39* belonged to ABF-type bZIP transcription factors, which are induced by ABA [36] and are critical in plant abiotic stresses tolerance [37,38]. Therefore, the results of our evolutionary relationship analysis imply that bZIP proteins in the group II might be associated with ABA. *GmbZIP9*, *GmbZIP10 and GmbZIP11* had high homology with *AtbZIP51,* which was a VIP1 type *bZIP* transcription factor and has been reported to impact plant growth and stress response and each were classified into the group II [39]. These results suggest that the 15 soybean bZIP family members are relevant for plant resistance.

The analysis of *cis*-acting promoter elements showed that the regions of the 15 soybean bZIP family members were rich in *cis*-acting elements related to stress responses, such as DRE, ABRE, MYB, MYC, LTR and TGACG-motif elements [40,41,42,43,44]. These results also indicate that the 15 soybean bZIP family members play important roles in stress response regulation. Based on recent research, a *bZIP* transcription factor of sweet potato, *IbbZIP37,* interacted with the ABRE *cis*-element to cope with drought, salt and heat shock stresses [45]. In another study, the *FtbZIP83* gene from Fagopyrum tataricum showed an increased adaptability to drought and salt stresses and its promoter had high activity in transgenic *Arabidopsis* [46], and stress responsive *cis*-acting elements, such as DRE, ABRE, MYB, MYC, LTR and TGACG-motif elements, were found in their promoter regions [45,46]. In our study, the same *cis*-acting elements were detected in the *GmbZIP2* promoter. qRT-PCR analysis also indicated that the expression of the 15 soybean bZIP family members could be induced by drought and salt stresses. We found that 14 soybean bZIP family members could be induced by drought, except for *GmbZIP1*. However, the expression levels of *GmbZIP8*, *GmbZIP9*, *GmbZIP10*, *GmbZIP11*, *GmbZIP12*, *GmbZIP13 and GmbZIP14* were nearly unchanged under salt stress, which suggests that these 15 soybean bZIP family members are involved in different stress signal pathways. *GmbZIP2* differed from the other soybean bZIP family members and had high expression levels both under drought and salt stresses, which suggests that *GmbZIP2* may be involved in multiple stress responses. Therefore, to further explore the function of the soybean bZIP family members under stress conditions, *GmbZIP2* was chosen to further analyze its expression patterns under drought and salt stresses. Its molecular characteristics and the expression pattern analyses demonstrated that GmbZIP2 is localized in the nucleus and can be induced by multiple stresses, which suggests that *GmbZIP2* plays a central role in plant resistance. Functional identification analyses showed that the overexpression of *GmbZIP2* in *Arabidopsis* and soybean hairy root systems enhanced plant tolerance to drought and salt stresses.

A recent study found that *OsbZIP73* overexpression in rice improved resistance to cold by modulating ABA levels and reactive oxygen species (ROS) homeostasis [29]. However, how *GmbZIP2* works in planta continues to attract our attention. Stress response genes such as *GmMYB48*, *GmWD40*, *GmDHN15*, *GmGST1 and GmLEA* were found to be upregulated in the transgenic soybean hairy root lines. *GmMYB48* encodes a MYB type transcription factor that was reported to contribute to the plant stress tolerance and promotes the expression of stress-responsive genes [47]. *GmWD40* encodes a WD40 structure protein that was reported to play crucial roles in diverse protein–protein interactions by acting as a scaffolding molecule and thus assisting in the proper functionality of proteins and being involved with abiotic stress response [48]. GmDHN15 is a dehydrin protein that serves a purpose in drought stress response [46]. GmLEA is a late-embryogenesis abundant protein with a major role in drought and other abiotic stresses tolerance in plants [49]. GmGST1 belongs to the glutathione S-transferase protein family that was reported to be involved in maintaining cell redox homeostasis and protecting organisms against oxidative stress under stress conditions [50,51,52]. The high expression levels of these stress responsive genes in transgenic *GmbZIP2* soybean hairy root plants indicated that *GmbZIP2* enhanced plant resistance by regulating the expression of numerous stress responsive genes. However, among these stress-responsive genes, *GST* encodes a ROS-scavenging enzyme and modulates ROS homeostasis, which suggests that *GmbZIP2* may modulate ROS homeostasis via GST in planta. Therefore, we conclude that soybean bZIP proteins play an important role in resisting drought and salt stresses.

## 4. Materials and Methods

### 4.1. De Novo Transcriptome Sequencing

Four-leaf stage untreated soybean seedlings were taken out of flowerpots, washed off with water and then the seedlings were dehydrated on filter paper and soaked in water with 150 mM NaCl for 2 h to prepare the samples for RNA-seq analysis. The detailed methods of RNA-seq were described by Yu et al., 2018 [53]. The data of de novo transcriptome sequencing is seen in Appendix A.

### 4.2. Identification of Soybean bZIP Proteins

All of the candidate *bZIP* genes were identified according to the data from the de novo transcriptome sequencing and the National Center for Biotechnology Information (NCBI) database. The gene sequences and protein sequences of *Arabidopsis* and soybean bZIPs were acquired from TAIR [54] and JGI Glyma1.0 [55] annotation, respectively.

### 4.3. Phylogenetic Analysis and Chromosome Localization of the 15 Soybean bZIPs

The positional information of the 15 soybean bZIPs was obtained from Phytozome. All bZIPs were located on the soybean chromosomes by using MapInspect software. Clustal X 2.0 was applied for protein sequence comparison analysis of the *Arabidopsis* and soybean bZIPs [56]. The protein sequences used the *Arabidopsis* and soybean references reported in References [17,32]. A phylogenetic tree was constructed using the adjacent method by MEGA6.0 with a 1000 bootstrap value [57].

### 4.4. Structure and Cis-Acting Element Analysis of the 15 Soybean bZIPs

The genome and protein sequences of the 15 soybean bZIPs were downloaded from Phytozome. The online software Gene Structure Display Server (http://gsds.cbi.pku.edu.cn/) was used to analyze the structure of 15 soybean *bZIP* genes [46]. The protein structure of the 15 soybean bZIPs were obtained by using ExPAsy-PROSITE (http://prosite.expasy.org/). We used the 2000 bp upstream of the start codon as the promoter for the 15 soybean bZIPs and all sequences were obtained from Phytozome. The *cis*-acting elements in these promoter sequences were analyzed with the online PlantCARE software (http://bioinformatics.psb.ugent.be/webtools/plantcare/html/). The data are available in Appendix A.

### 4.5. Tissue-Specific Expression Analysis of 15 Soybean bZIPs

The expression pattern data of the 15 *bZIP* family members in 26 different tissues of soybean (flower open, flower un-open, lateral root standard, leaf ammonia, leaf nitrate, leaf standard, leaf symbiotic condition, leaf urea, nodules symbiotic condition, root tip standard, root ammonia, root nitrate, root standard, root symbiotic condition, root urea, shoot tip standard, stem standard, flower, leaves, nodules, pod, root, root hairs, seed, shoot apical meristem and stem organs) under normal conditions were downloaded from Phytozome (https://phytozome.jgi.doe.gov/pz/portal.html) and then Hierarchical clustering was performed with Heml1.0 software (http://www.patrick-wied.at/static/heatmapjs/). The relevant data are listed in Appendix A.

### 4.6. qRT-PCR Analysis

Four-leaf stage soybean seedlings grown under normal conditions (22 °C, light 16 h/dark 8 h, 50% of relative humidity) were taken out of their flowerpots, washed off with water, dehydrated on filter paper and soaked in 150 mM NaCl and then samples were taken at 0, 3, 6 and 12 h, respectively and saved at −70 °C after freezing in liquid nitrogen. The total RNA was isolated, and qRT-PCR was performed according to Yu et al. (2018) [53]. The primers for qRT-PCR of the 15 soybean *bZIP* transcription factor genes were designed by using the Primer 5.0 software (Premier, Ottawa, Canada) (Appendix A).

### 4.7. Subcellular Localization of GmbZIP2

The ORF of *GmbZIP2* without the stop codon was fused to the N-terminus of the hGFP gene under the control of the constitutive CaMV 35S promoter using the specific primers for GmbZIP2-hGFP (In-Fusion^®^ HD Cloning Kit, Takara). The transient expression of the GmbZIP2-hGFP fusion construct and the hGFP control vector in *Arabidopsis* protoplast cells were performed as in Liu et al. (2013) [58]. After being dark cultured for 12 h at 25 °C, the *Arabidopsis* protoplast cells were observed with a confocal laser scanning microscope (LSM700, CarlZeiss, Oberkochen, Germany) [59]. *GmbZIP2* was inserted into the prokaryotic expression vector pCold (TaKaRa) by using a pair of specific primers (Appendix A). The protein of the plant nucleus, cytoplasm and cytomembrane were isolated by using the plant nuclear cytoplasm and cytomembrane extraction kit (Best Bio, Shanghai, China) and then the plant nucleus protein was detected by western blot analysis with anti-GFP.

### 4.8. Expression Pattern Analysis of GmbZIP2 under Different Stresses

Soybean seedlings were cultured at normal conditions (22 °C, light 16 h/dark 8 h, 50% of relative humidity) until the four-leaf stage, then were treated under four different stresses, including heat (45 °C for 12 h), 150 μM ABA, 200 mM mannitol and cold (4 °C for 12 h). Samples of the soybean seedlings under each treatment were obtained at 0, 1, 2, 4, 6 and 12 h, respectively. The total RNA was isolated, and qRT-PCR was performed according to Yu et al. (2018) [53]. The primers for qRT-PCR of *GmbZIP2* were designed by using the Primer 5.0 software (Appendix A).

### 4.9. Effect of Drought and Salt Stresses on Transgenic Arabidopsis Growth

*GmbZIP2* was cloned into pCAMBIA1302 and driven by the CaMV 35S promoter. The *GmbZIP2* gene was transformed into *Arabidopsis* (*Columbia-0*, WT) by the floral dip method [60]. The transgenic plants were first screened on half MS medium supplemented with 50 mg L^−1^ hygromycin. Seeds from each T1 plant were individually collected. Selected T2 plants were propagated. T3 progeny homozygotes were obtained for further analysis.

To investigate the effect of drought stress on the growth of transgenic *Arabidopsis*, seeds of the three T3 homozygous transgenic lines and wild-type line were sown on MS agar plates for germination, kept at 4 °C for 3 days and then transferred to normal conditions (22 °C, light 16 h/dark 8 h, 50% of relative humidity) to continue to grow. After 14 days of growth, the seedlings from each line were carefully transferred to flowerpots containing vermiculite and nutrient soil (*v/v* = 1:1) for 10 days of growth and then the seedlings were used in the phenotyping experiment. For drought treatment, water was withheld for 2 weeks and then followed by a full re-watering and recovery period. Two weeks later, the survival rate was calculated. For salt treatment, the seedlings were exposed to 200 mM NaCl stress for a week and the survival rate was calculated at the end of the treatment. The treated and untreated *Arabidopsis* seedlings with drought and NaCl were collected for RNA preparation and physiological and biochemical determination. The measurement of chlorophyll and MDA contents were carried out as described by Cui and Wang et al. (2006) [61] and Abdallah et al. (2007) [62] and the determination of REC was carried out as in Sharp et al. (1990) [63].

### 4.10. Hairy Root Induction of Soybean Transformation under Drought and Salt Stresses

Transgenic soybean hairy root composite plants were generated by the method described by Shi et al. (2018) [64]. The water-deficient treatment was performed for 2 weeks and then transgenic soybean *35S::GmbZIP2* and EV-Control seedlings were returned to normal growth conditions for one week [65]. Under salt stress, transgenic soybean seedlings and controls were gown in 200 mM NaCl for a week. Ten soybean seeds were put in each pot and three pots formed a group. Each stress treatment experiment was repeated 3 times. Similarly, the treated and untreated soybean hairy roots were collected for RNA isolation and physiological and biochemical experimentation. The measurements of chlorophyll, MDA and proline contents were carried out as described by Cui and Wang et al. (2006) [61] and Abdallah et al. (2007) [62] and the determination of REC was carried out as in Sharp et al. (1990) [63].

### 4.11. Statistical Analyses

Each assay was performed at least three times independently to ensure the reliability of our results. Vertical bars indicate ± SE of three replicates and statistical comparison among groups was conducted via a Student’s *t*-test. The one star indicates differences in comparison with the control lines at *p* < 0.05, the double star indicates highly significant differences in comparison with the control lines at *p* < 0.01.

## 5. Conclusions

In this study, we identified 15 soybean bZIPs that were upregulated in response to drought stress from the data of de novo transcriptome sequencing of soybean. The 15 soybean *bZIP* transcription factor family members were phylogenetically divided into six groups and distributed over 12 chromosomes. Afterwards we found that the 15 soybean bZIPs genes were differentially expressed under drought and salt stresses. *GmbZIP2* was induced significantly by drought and salt stresses. The results of phenotyping assays show that overexpressing *GmbZIP2* in *Arabidopsis* and soybean conferred an enhanced tolerance to drought and salt. qRT-PCR revealed an elevated transcription of *GmMYB48*, *GmWD40*, *GmDHN15*, *GmGST1* and *GmLEA* in transgenic soybean overexpressing *GmbZIP2* and thus *GmbZIP2* was demonstrated to be involved in plants’ tolerance to drought and salt stresses. These results provide vital clues to understanding the regulatory mechanism of *bZIP* genes in abiotic stress responses in plants.

## Figures and Tables

**Figure 1 ijms-21-00670-f001:**
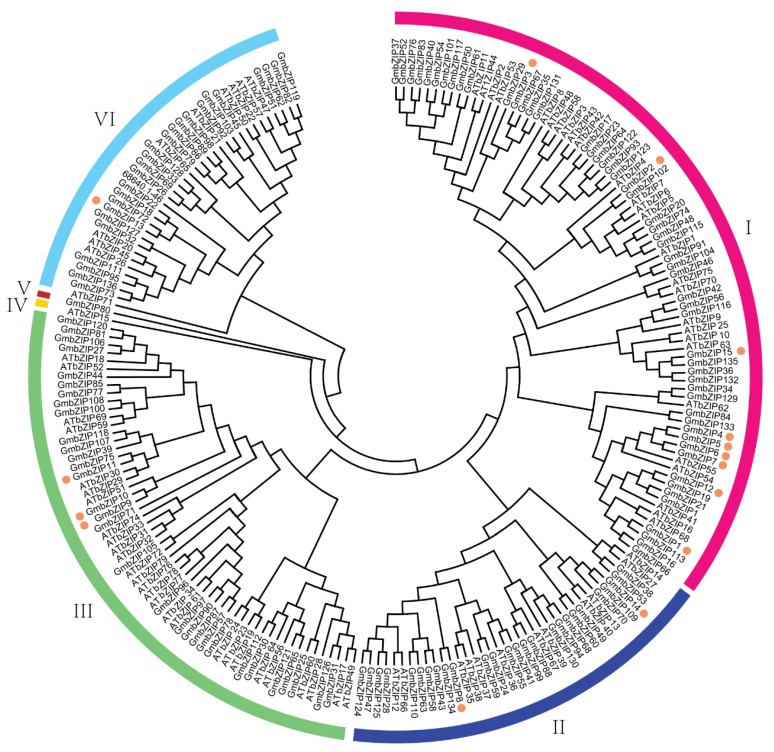
Phylogenetic relationships of bZIPs with soybean and *Arabidopsis thaliana*. The phylogenetic tree was produced by MEGA 5.0 software based on the comparison of amino acid sequences of GmbZIPs. The neighbor-joining method was used and the bootstrap values were set at 1000. The brown dot means *GmbZIP*s that were high induced in transcriptome data (Appendix A). *GmbZIP*s were divided into six classes.

**Figure 2 ijms-21-00670-f002:**
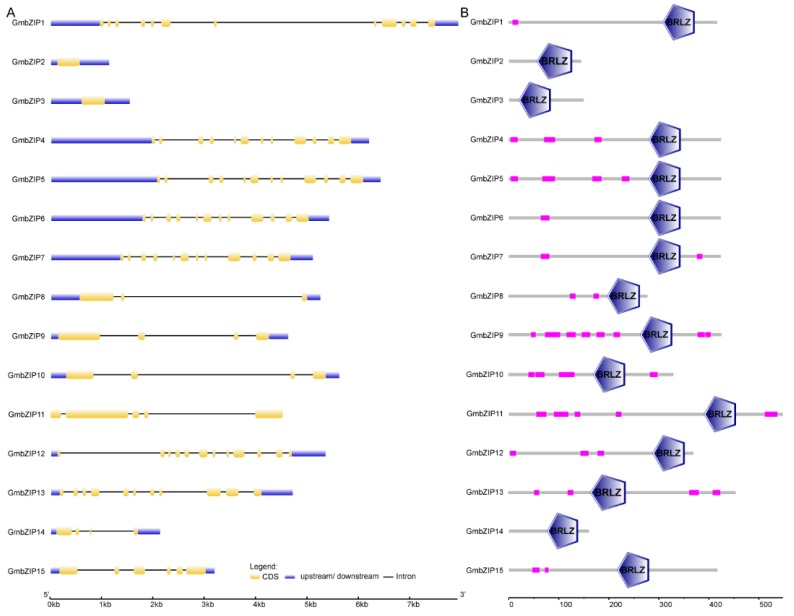
Gene structure analysis of soybean bZIPs. (**A**) Intron–exon structures of 15 *soybean bZIP* genes. (**B**) Protein structure analysis of 15 soybean *bZIP* genes.

**Figure 3 ijms-21-00670-f003:**
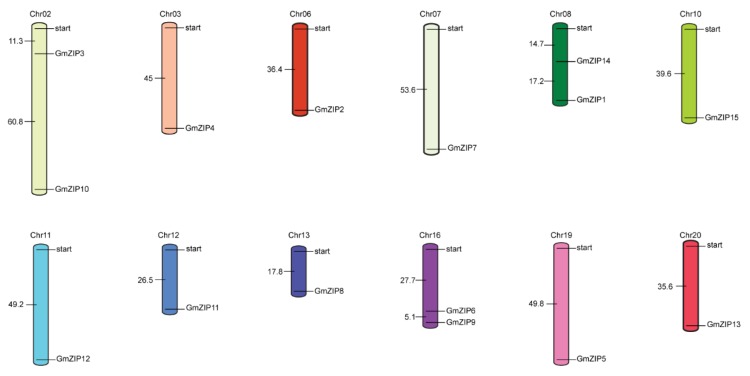
Distribution of the *GmbZIP* genes in soybean genome. The 15 soybean *bZIP* genes distributed on the 12 chromosomes.

**Figure 4 ijms-21-00670-f004:**
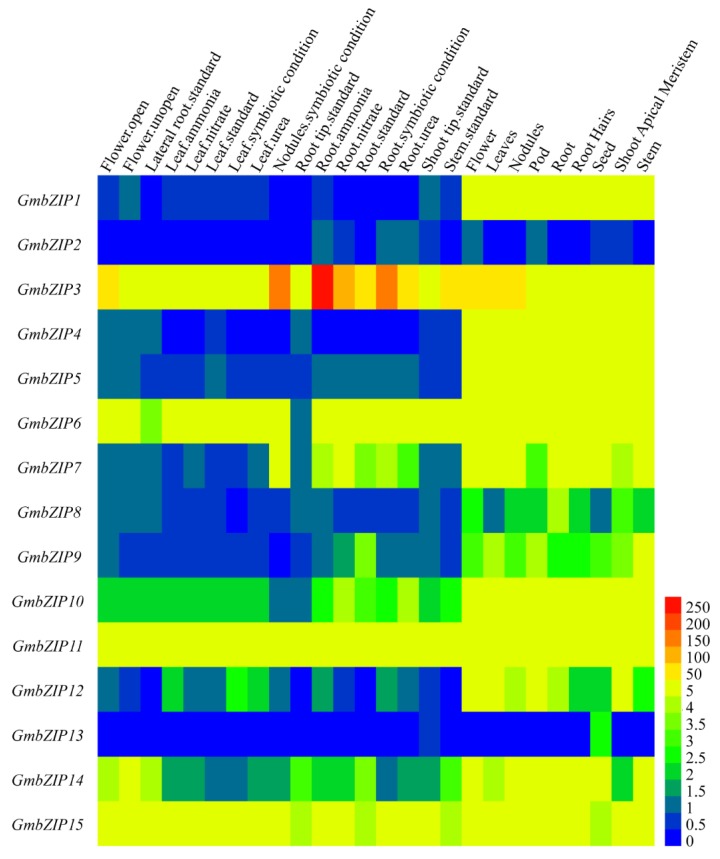
The tissue specific expression patterns of the 15 *GmbZIP* genes. The tissues of soybean from left to right are flower open, flower unopen, lateral root standard, leaf ammonia, leaf nitrate, leaf standard, leaf symbiotic condition, leaf urea, nodules symbiotic condition, root tip standard, root ammonia, root nitrate, root standard, root symbiotic condition, root urea, shoot tip standard, stem standard, flower, leaves, nodules, pod, root, root hairs, seed, shoot apical meristem and stem organs. The color legend refers to the different expression levels under normal conditions.

**Figure 5 ijms-21-00670-f005:**
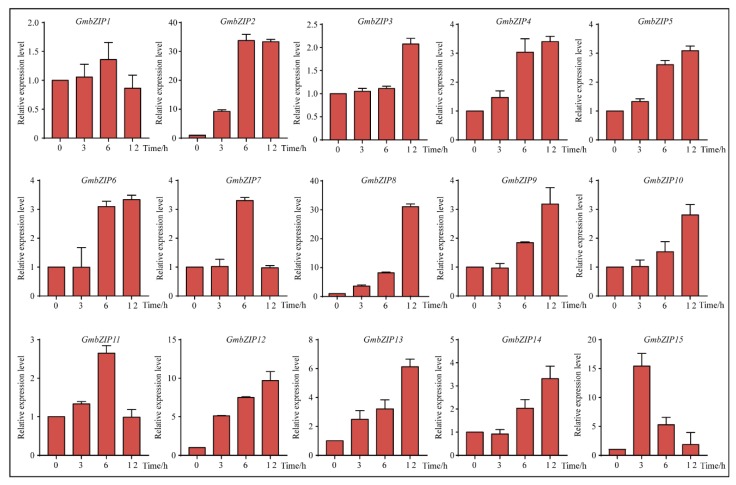
Expression patterns of the 15 *GmbZIP* genes under drought treatment. Quantitative real-time polymerase chain reaction (qRT-PCR) data are normalized using soybean *Actin* (*U60506*) and displayed as relative to 0 h. The X-axes show time periods and y-axes depict scales of relative expression level (error bars indicate SD).

**Figure 6 ijms-21-00670-f006:**
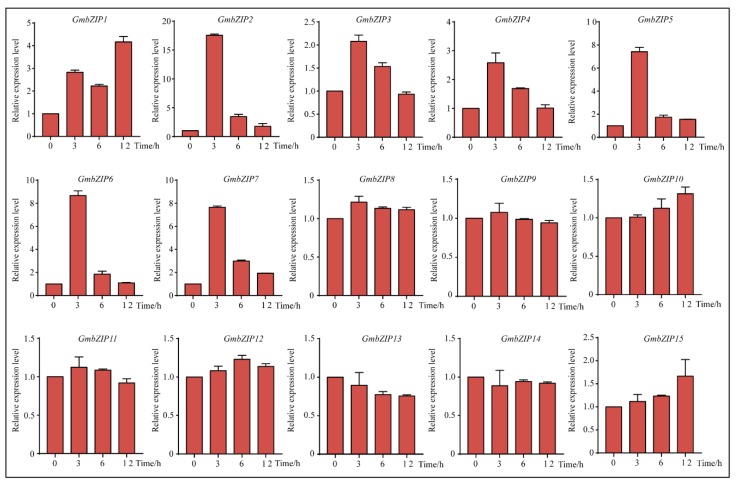
Expression patterns of the 15 *GmbZIP* genes under salt treatment. qRT-PCR data are normalized using soybean *Actin* (*U60506*) and displayed as relative to 0 h. The X-axes show time periods and y-axes depict scales of relative expression level (error bars indicate SD).

**Figure 7 ijms-21-00670-f007:**
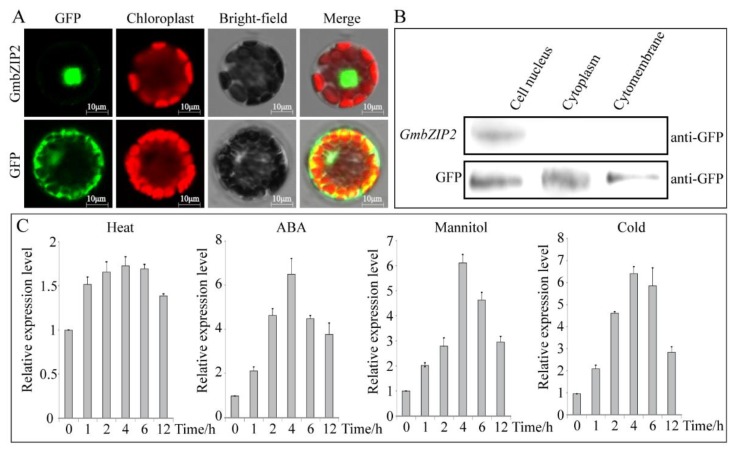
GmbZIP2 subcellular localization and expression patterns under different stresses. (**A**) Subcellular localization analysis of GmbZIP2 protein. The scale bar was shown by 10 μm. (**B**) Western blot detection analysis of GmbZIP2 protein subcellular localization. (**C**) The expression patterns of *GmbZIP2* under different stresses. Vertical bars in C indicate ± SE of three replicates.

**Figure 8 ijms-21-00670-f008:**
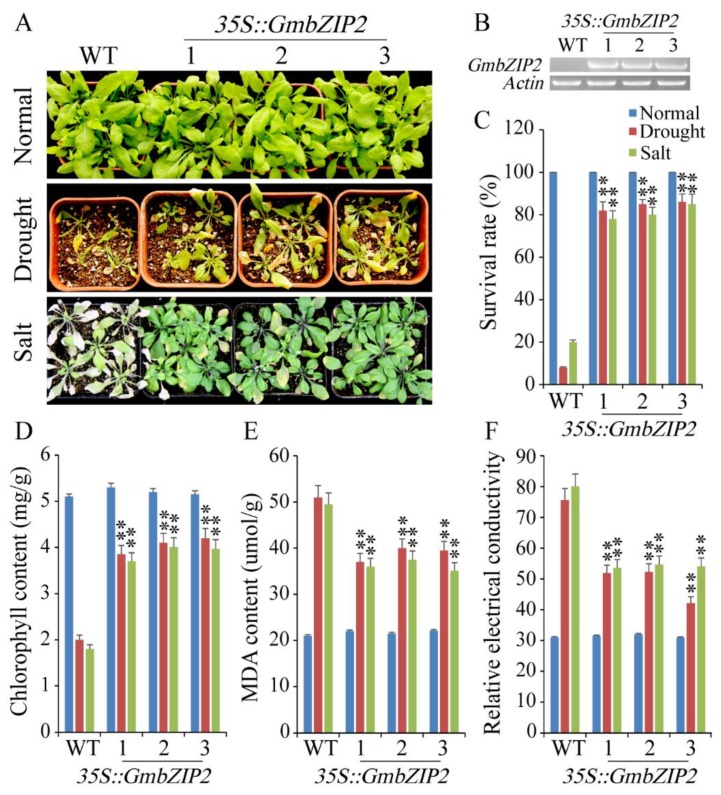
Phenotype identification of transgenic and WT (wild-type, WT, Col-0) seedlings under drought and salt stresses. (**A**) The growth states of transgenic and WT seedlings under drought and salt stresses. (**B**) The transcription levels of *GmbZIP2* in transgenic and WT seedlings by semi-quantitative PCR. (**C**–**F**) The survival rates, Chlorophyll content, malondialdehyde (MDA) content and REC (relative electrical conductivity) of transgenic and WT *Arabidopsis* seedlings under drought and salt stresses, respectively. Vertical bars in C–F indicate ± SE of three replicates. ** indicate significant differences in comparison with the WT lines at *p* < 0.01.

**Figure 9 ijms-21-00670-f009:**
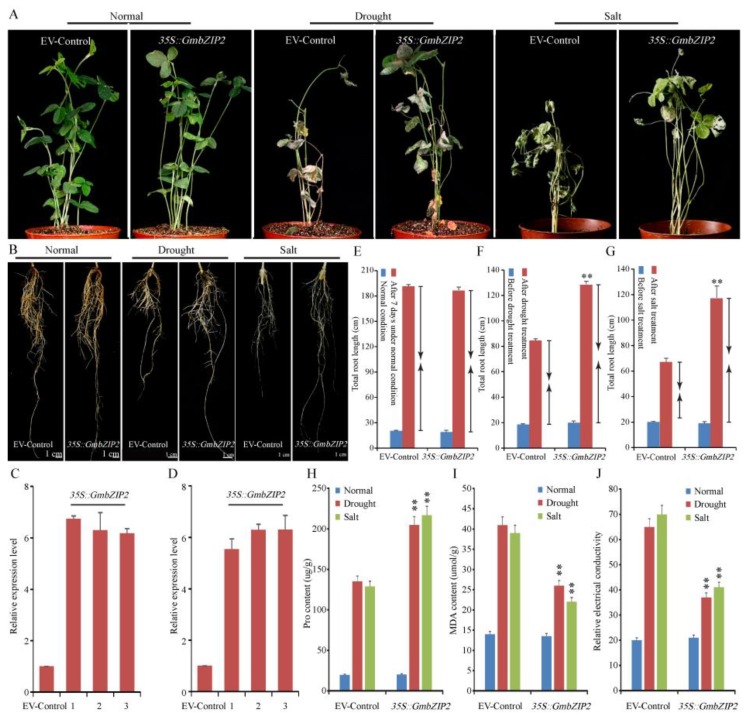
Phenotype and function identification of soybean *GmbZIP2* under drought and salt stresses. (**A**,**B**) Growth states of transgenic soybean hairy root composite plants and EV-control (the empty plasmid of pCAMBIA3301) plants under drought and salt stresses. (**C**,**D**) Transcriptional level of *GmbZIP2* in transgenic soybean hairy roots. (**E**) The root elongation of transgenic soybean hairy root composite plants and EV-control plants under non-stress condition. (**F**) Root elongation of transgenic soybean hairy root composite plants and EV-control plants under drought condition. (**G**) Root elongation of transgenic soybean hairy root composite plants and EV-control plants under salt condition. (**H**) Proline content of transgenic soybean hairy root composite plants and EV-control plants under drought and salt conditions. (**I**) MDA content of transgenic soybean hairy root composite plants and EV-control plants under drought and salt conditions. (**J**) Relative electrical conductivity of transgenic soybean hairy root composite plants and EV-control plants under drought and salt conditions. Vertical bars indicate ± SE of three replicates. ** indicates significant differences in comparison with the control lines at *p* < 0.01.

**Figure 10 ijms-21-00670-f010:**
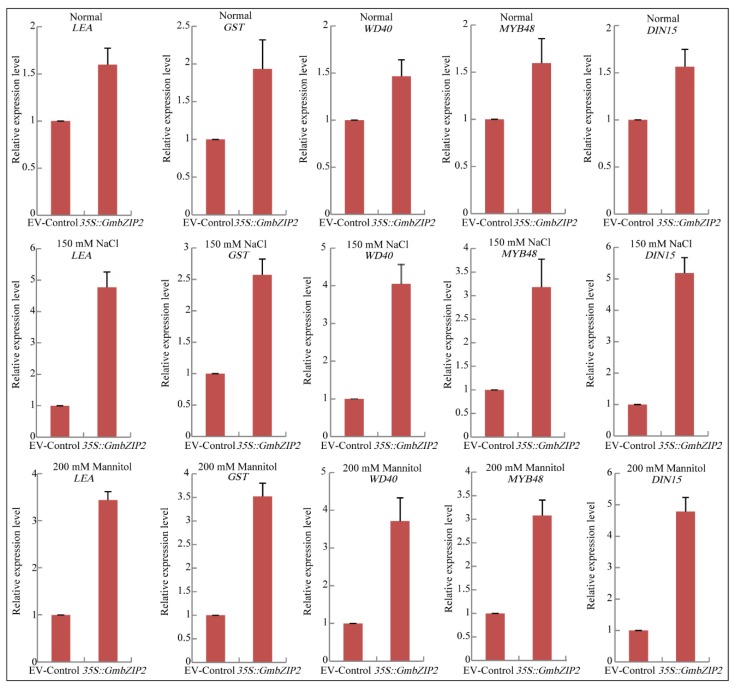
Expressions of five stress-responsive genes in transgenic *GmbZIP2* soybean hairy root plants under normal conditions, 150 mM NaCl and 200 mM mannitol treatments detected by qRT-PCR. Vertical bars indicate ± SE of three replicates.

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
