# Peer review of "The Soybean bZIP Transcription Factor Gene GmbZIP2 Confers Drought and Salt Resistances in Transgenic Plants"

_ijms, 2020, doi:10.3390/ijms21020670_

Round 1

Reviewer 1 Report

The authors identified differentially expressing bZIP genes under drought and salt stress in soybean and attempted to characterize one of them (which showed the highest levels of stress-induction) by generating transgenic plants.

The authors must consider the serious editing of the poorly written manuscript. Furthermore, 'normal' may be replaced with 'control'/'mock treatment'; 'hairy root complex' by 'hairy root composite' plants. The methodology should be improved as well. The bioinformatic analyses of the bZIP transcription factor genes and the conclusions drawn provide no new information. It should be toned down or moved to supplement.

3. Rather than investigating differential expression of some known stress-responsive genes namely GmMYB48, GmWD40, GmDHN15, GmGST1, and GmLEA in the transgenic hairy roots, the authors should have analyzed the transcriptome of the transgenic versus wild type plants which would identify the downstream genes of the GmbZIP2 transcription factor, thereby correctly addressing the molecular mechanisms behind the differential growth rates of WT vs overexpressors under stress. 

Author Response

The authors identified differentially expressing bZIP genes under drought and salt stress in soybean and attempted to characterize one of them (which showed the highest levels of stress-induction) by generating transgenic plants.

The authors must consider the serious editing of the poorly written manuscript. Furthermore, 'normal' may be replaced with 'control'/'mock treatment'; 'hairy root complex' by 'hairy root composite' plants. The methodology should be improved as well. Response:

Dear reviewer, it is really true as reviewer suggested that we have amended the word and re- written the method of this part. Transgenic soybean hairy root composite plants were generated by method of hairy root induction it was the A. rhizogenes strain K599 (NCPPB2659), carrying the recombinant plasmid of GmbZIP2-pCAMBIA3301 (OE-GmbZIP2) and the empty plasmid of pCAMBIA3301 (EV-Control) which were both driven by CaMV35S promoter, they were introduced into cotyledonary node and/or hypocotyl of the 7-day-old soybean (Williams 82) which were cultivated in normal condition (22°C, light 16 h/dark 8 h, 50% of relative humidity) for drought and salt stresses assay. After the injection, the OE-GmbZIP2 and EV-Control plants were covered plastic cups to maintain humidity. During the plant generated new roots (about 2 weeks) provided nutritious soil in time to ensure infection site buried in the soil to grow new roots. Then took the upper part of the inoculation site and transplanted hairy roots of the soybean seedlings in new nutritive soil were cultured for 5 days. For drought treatment, the water-deficient treatment was performed for 2 weeks, and then transgenic soybean 35S::GmbZIP2 and EV-Control seedlings were returned to normal growth conditions for one week. Under salt stress, transgenic soybean seedlings and Control were gown in 200 mM NaCl for a week. The transgenic soybean seedlings and control were untreated with drought and salt as control group.

The bioinformatic analyses of the bZIP transcription factor genes and the conclusions drawn provide no new information. It should be toned down or moved to supplement.

Response:

According to reviewer comments, the bioinformatic analyses date of the bZIP transcription factor genes were provided in supplement

Rather than investigating differential expression of some known stress-responsive genes namely GmMYB48, GmWD40, GmDHN15, GmGST1, and GmLEA in the transgenic hairy roots, the authors should have analyzed the transcriptome of the transgenic versus wild type plants which would identify the downstream genes of the GmbZIP2 transcription factor, thereby correctly addressing the molecular mechanisms behind the differential growth rates of WT vs overexpressors under stress. Dear reviewer, the stress response genes GmMYB48, GmWD40, GmDHN15, GmGST1, and GmLEA were found to be up-regulated in the de novo transcriptome date of soybean under drought stress (Figure S6). In our subsequent operations, we verified that their transcription level were increased in transgenic soybean hairy root lines (Figure 10), and these genes were reported in keeping plant from succumbing to stressful environments by regulate diverse plant processes including the osmotic balance and ROS homeostasis and so on [1-7]. Then we have discovered a plurality of cis-acting element that could be bound by bZIP transcription factor in 5 stress response genes promoter (Figure S7). So we speculate that these stress response genes possibly are downstream genes of the GmbZIP2 transcription factor. Since time is limited, this survey is not comprehensive, we feel regret that we lack of identify the downstream genes of the GmbZIP2 transcription factor, but we certain that we could do our best efforts to figure it out by conducting dual-luciferase reporter gene assay. In short, GmbZIP2 may be involved in regulation of these genes expression and modulate directly or indirectly ROS homeostasis via GST in planta to enhance stress resistance of plants.

Supplemental Table 6. RNA-seq data analyses of 5 stress response genes.  Response:

Gene

Gene ID

CK_treat-Expression

GH_treat-Expression

log2FoldChange(GH_treat/CK_treat)

Up/Down-Regulation

GmDHN15

Glyma.11G149900

3.62041432

46354.88544

13.64427896

up

GmWD40

Glyma.15G129100

1.33916319

3267.956744

11.25284139

up

GmMYB48

Glyma.15G041100

23.43817295

3904.360652

7.380082399

up

GmGST1

Glyma.07G139800

39.2812351

31508.76997

9.647697427

up

GmLEA

Glyma.U018200

0.34561815

82.41961276

7.897664872

up

Supplemental Table 7. The number of cis-acting elements that could be bound by basic leucine zipper (bZIP) transcription factors in 5 stress response genes promoter.

Gene

ACGT cis-acting elements

GmDHN15

21

GmWD40

12

GmMYB48

16

GmGST1

8

GmLEA

10

Guo, X.; Zhang, L.; Wang, X.; Zhang, M.; Xi, Y.; Wang, A.; Zhu, J. Overexpression of Saussurea involucrata dehydrin gene SiDHN promotes cold and drought tolerance in transgenic tomato plants. PLoS One. 2019, 14, e0225090. Liao, Y, Zou, H.F.; Wang, H.W.; Zhang, W.K.; Ma, B.; Zhang, J.S, Chen, S.Y. Soybean GmMYB76, GmMYB92, and GmMYB177 genes confer stress tolerance in transgenic Arabidopsis plants. Cell Res. 2008, 18, 1047-1060. Mishra, A.K.; Muthamilarasan, M.; Khan, Y.; Parida, S.K.; Prasad, M. Genome-wide investigation and expression analyses of WD40 protein family in the model plant foxtail millet (Setaria italica L.). PLoS One2014, 9, e86852. Magwanga, R.O.; Lu, P.; Kirungu, J.N.; Lu, H.; Wang, X.; Cai, X.; Zhou, Z.; Zhang, Z.; Salih, H.; Wang, K.; Liu, F. Characterization of the late embryogenesis abundant (LEA) proteins family and their role in drought stress tolerance in upland cotton. BMC Genet. 2018, 19, 6. Jha, B.; Sharma, A.; Mishra, A. Expression of SbGSTU (tau class glutathione S-transferase) gene isolated from Salicornia brachiate in tobacco for salt tolerance. Mol Biol Rep2011, 38, 4823–4832. Rong, W.; Qi L.; Wang, A.; Ye, X.; Du, L.; Liang, H.; Xin Z.; Zhang, Z. The ERF transcription factor TaERF3 promotes tolerance to salt and drought stresses in wheat. Plant Biotechnol J2014, 12, 468–479. Qi, J.; Song, C.P.; Wang, B.; Zhou, J.; Kangasjarvi, J.; Zhu, J.K.; Gong, Z. Reactive oxygen species signaling and stomatal movement in plant responses to drought stress and pathogen attack. J Integr Plant Biol2018, 60, 805–826

Dear reviewer,

Special thanks to you for your good comments.

We tried our best to improve the manuscript and made some changes in the manuscript.  These changes will not influence the content and framework of the paper. And here we did not list the changes but marked in red in revised paper. We appreciate for Editors/Reviewers’ warm work earnestly, and hope that the correction will meet with approval. Once again, thank you very much for your comments and suggestions.

Best regards!

Yours sincerely,

Yan Yang E-mail:[email protected]

Name: Zhao-shi Xu E-mail: [email protected]

Reviewer 2 Report

In this manuscript the Authors report the identification of 15 bZIP transcription factors using de novo RNA sequencing from drought-induced soybean seedlings. The location of these 15 genes on chromosomes was determined and phylogenetic analysis performed as well as analysis of gene and protein structure. Analysis of promotors have shown that these 15 genes contain cis-acting elements involved in response to abiotic stresses. Expression study showed that the transcript level of all 15 bZIP increased in response to salt stress and drought, but the GmbZIP2 showed the highest increase. Therefore the Authors generated the transgenic Arabidopsis and soybean plants producing GmbZIP2 and have shown that overexpression of GmbZIP2 in Arabidopsis and soybean hairy roots enhanced plants resistance to drought and salt stress. The Authors have shown that the improved salt and drought tolerance in GmbZIP2 -transgenic soybean hairy root could be due to an upregulation of genes involved in plant stress tolerance, increase proline and decrease malondialdehyde (MDA) and relative electrical conductivity (REC) content. In Arabidopsis GmbZIP2 can enhance salt  and drought tolerance by preventing chlorophyll degradation and reducing MDA and REC content.

There are several items that could be improved.

The phylogenetic analysis of bZIPs, distribution of bZIPs on chromosomes, the exon-intron structure and analysis of cis-acting element in promoters of all GmbZIP were already reported by Zhang et al. BMC Genomics (2018) 19:159 (not cited in this manuscript) and/or by Wang et al. BMC Genomics (2015) 16:1053. Moreover, the expression profiles of all identified GmbZIP in different tissues and under drought and flooding in soybean leaves was reported in the paper Zhang et al. BMC Genomics (2018). Therefore , first, the paper of Zhang et al. BMC Genomics (2018) 19:159 should be referenced. Secondly, could the Authors discuss their results with those published by Zhang et al. (2018) and Wand et al. (2015).Thirdly, it would be beneficial  to include information which GmbZIPs in both the above mentioned papers correspond to GmbZIP identified in the currently reviewed paper (information can be included in Supplementary Materials).

A total of 160 and 138 bZIP genes were identified from soybean genome by Zang et al. (2018) and Wang et al. (2015), respectively. In this manuscript 136 bZIP genes were identified. In this manuscripts 15 up-regulated soybean bZIP were distributed among 6 groups (Figure SI), but in the Figure 1 all soybean bZIP were also divided into 6 groups. In the paper of Zhang et. al. (2018) all soybean bZIP were divided into 12 groups. Could the Author explained these differences?

There also are some issues about the experimental protocol that should be clarified.

For example:

It is not clear how the transgenic GmbZIP2 soybean hairy root plants and were cultivated and subjected to stresses (drought, NaCl, mannitol), in soil or hydroponically? How old were the plants? Figure 8 shows the results of study the phenotype of transgenic Arabidopsis plants expressing GmbZIP2. How old were the plants? Seedlings or leaves were taken for the study of chlorophyll, MDA and REC content? It is not clear.    Figure 5 and Figure 6 show expression profiles of 15 GmbZIP genes under drought and salt stress, respectively. I do not fully understand whether the seedlings were taken for testing or whether they were already leaves, given the age of the plant. There are some inaccuracies in the description of Statistical Analysis in Materials and Methods and descriptions of Figure legends.

In conclusion, the manuscript is suitable to be published in International Journal of Molecular Sciences, but after correction.

Author Response

In this manuscript the Authors report the identification of 15 bZIP transcription factors using de novo RNA sequencing from drought-induced soybean seedlings. The location of these 15 genes on chromosomes was determined and phylogenetic analysis performed as well as analysis of gene and protein structure. Analysis of promotors have shown that these 15 genes contain cis-acting elements involved in response to abiotic stresses. Expression study showed that the transcript level of all 15 bZIP increased in response to salt stress and drought, but the GmbZIP2 showed the highest increase. Therefore the Authors generated the transgenic Arabidopsis and soybean plants producing GmbZIP2 and have shown that overexpression of GmbZIP2 in Arabidopsis and soybean hairy roots enhanced plants resistance to drought and salt stress. The Authors have shown that the improved salt and drought tolerance in GmbZIP2 -transgenic soybean hairy root could be due to an upregulation of genes involved in plant stress tolerance, increase proline and decrease malondialdehyde (MDA) and relative electrical conductivity (REC) content. In Arabidopsis GmbZIP2 can enhance salt and drought tolerance by preventing chlorophyll degradation and reducing MDA and REC content.

There are several items that could be improved.

The phylogenetic analysis of bZIPs, distribution of bZIPs on chromosomes, the exon-intron structure and analysis of cis-acting element in promoters of all GmbZIP were already reported by Zhang et al. BMC Genomics (2018) 19:159 (not cited in this manuscript) and/or by Wang et al. BMC Genomics (2015) 16:1053. Moreover, the expression profiles of all identified GmbZIP in different tissues and under drought and flooding in soybean leaves was reported in the paper Zhang et al. BMC Genomics (2018).

Therefore, first, the paper of Zhang et al. BMC Genomics (2018) 19:159 should be referenced.

Secondly, could the Authors discuss their results with those published by Zhang et al. (2018) and Wand et al. (2015).

Thirdly, it would be beneficial to include information which GmbZIPs in both the above mentioned papers correspond to GmbZIP identified in the currently reviewed paper (information can be included in Supplementary Materials).

Response:

Dear reviewer, firstly, we are very sorry for our negligence of not cited reference of Zhang et al. (2018) in this manuscript, and we have added this reference to manuscript. In previous study, the article (Zhang et al. 2018) have identified 160 bZIP genes in soybean and presented most of them may play more dominantly role in regulating drought stress. Our research towards this direction, 15 up-regulated soybean bZIPs in response to salt and drought stresses be verified by expression study, and the GmbZIP2 showed the highest increase treatments. Interesting here, we chose 5 stress response genes (GmMYB48, GmWD40, GmDHN15, GmGST1, and GmLEA) that up-regulated in the de novo transcriptome date of soybean under drought treatment, which reported that could protect plants against various abiotic stress. It may be could explain that the molecular mechanisms of GmbZIP2 gene enhance the resistance of plant by regulating the expression of numerous stress responsive genes, especially, modulation of ROS homeostasis aspect. Our conclusion of the soybean bZIP transcription factor gene GmbZIP2 confers drought and salt resistances in transgenic plants could be used as a reference for excavation excellent genes. Last, according to reviewer comments, the bioinformatic analyses date of the bZIP transcription factor genes were provided in supplement.

A total of 160 and 138 bZIP genes were identified from soybean genome by Zang et al. (2018) and Wang et al. (2015), respectively. In this manuscript 136 bZIP genes were identified. In this manuscripts 15 up-regulated soybean bZIP were distributed among 6 groups (Figure SI), but in the Figure 1 all soybean bZIP were also divided into 6 groups. In the paper of Zhang et. al. (2018) all soybean bZIP were divided into 12 groups. Could the Author explained these differences? There also are some issues about the experimental protocol that should be clarified.

Response:

Dear reviewer, the differences between 12 and 6 groups we have wrote in discussion section. In previous works published by Zang et al. (2018) and Wang et al. (2015), bZIP family members are classified into 12 groups in Arabidopsis and soybean according to their conserved DNA-binding domains (DBDs), However, here we re-divided the bZIP family members of Arabidopsis and soybean into 6 groups according to their conserved amino acid sequences. Considering the reviewer’s suggestion,we will further improve the method.

For example:

It is not clear how the transgenic GmbZIP2 soybean hairy root plants and were cultivated and subjected to stresses (drought, NaCl, mannitol), in soil or hydroponically? How old were the plants? Figure 8 shows the results of study the phenotype of transgenic Arabidopsis plants expressing GmbZIP2. How old were the plants? Seedlings or leaves were taken for the study of chlorophyll, MDA and REC content? It is not clear.    Figure 5 and Figure 6 show expression profiles of 15 GmbZIP genes under drought and salt stress, respectively. I do not fully understand whether the seedlings were taken for testing or whether they were already leaves, given the age of the plant. There are some inaccuracies in the description of Statistical Analysis in Materials and Methods and descriptions of Figure legends.

In conclusion, the manuscript is suitable to be published in International Journal of Molecular Sciences, but after correction.

It is not clear how the transgenic GmbZIP2 soybean hairy root plants and were cultivated and subjected to stresses (drought, NaCl, mannitol), in soil or hydroponically? How old were the plants?

Response:

As reviewer suggested that the detail methods of “transgenic GmbZIP2 soybean hairy root......” were refer to Shi et al., (2018), we elaborated as follows

Transgenic soybean hairy root composite plants were generated by method of hairy root induction it was the A. rhizogenes strain K599 (NCPPB2659), carrying the recombinant plasmid of GmbZIP2-pCAMBIA3301 (OE-GmbZIP2) and the empty plasmid of pCAMBIA3301 (EV-Control) which were both driven by CaMV35S promoter, they were introduced into cotyledonary node and/or hypocotyl of the 7-day-old soybean (Williams 82) which were cultivated in normal condition (22°C, light 16 h/dark 8 h, 50% of relative humidity) for drought and salt stresses assay. After the injection, the OE-GmbZIP2 and EV-Control plants were covered plastic cups to maintain humidity. During the plant generated new roots (about 2 weeks) provided nutritious soil in time to ensure infection site buried in the soil to grow new roots. Then took the upper part of the inoculation site and transplanted hairy roots of the soybean seedlings in new nutritive soil were cultured for 5 days. For drought treatment, the water-deficient treatment was performed for 2 weeks, and then transgenic soybean 35S::GmbZIP2 and EV-Control seedlings were returned to normal growth conditions for one week. Under salt stress, transgenic soybean seedlings and Control were gown in 200 mM NaCl for a week. The transgenic soybean seedlings and control were untreated with drought and salt as control group. 

Figure 8 shows the results of study the phenotype of transgenic Arabidopsis plants expressing GmbZIP2. How old were the plants? Seedlings or leaves were taken for the study of chlorophyll, MDA and REC content? It is not clear.

Response:

Dear reviewer, we described in our materials and methods in detail, to investigate the effect of drought stress on the growth of transgenic Arabidopsis, seeds of the three T3 homozygous transgenic lines and wild-type line were sowed on MS agar plates for germination, kept at 4°C for 3 days, and then transferred to normal conditions (22°C, light 16 h/dark 8 h, 50% of relative humidity) to continue to grow. After 14 days of growth, the seedlings from each line were carefully transferred to flowerpots containing vermiculite and nutrient soil (v/v=1:1) for 10 days of growth, and then the seedlings were used in the phenotyping experiment. For drought treatment, water was withheld for 2 weeks, and then followed by a full re-watering and recovery period. Two weeks later, survival rate was calculated. For salt treatment, the seedlings were exposed to 200 mM NaCl stress for a week, and the survival rate was calculated at the end of the treatment. The treated and untreated Arabidopsis seedlings with drought and NaCl were collected for RNA preparation and physiological and biochemical determination.

Figure 5 and Figure 6 show expression profiles of 15 GmbZIP genes under drought and salt stress, respectively. I do not fully understand whether the seedlings were taken for testing or whether they were already leaves, given the age of the plant.

Response:

Dear reviewer, let me explain that four-leaf stage treated with drought and salt soybean seedlings were collected for establishment of expression profiles to further validate the accuracy of the transcriptome data.

There are some inaccuracies in the description of Statistical Analysis in Materials and Methods and descriptions of Figure legends.

Response:

We have re-checked and made correction according to the reviewer’s comments.

Dear reviewer,

Special thanks to you for your good comments.

We tried our best to improve the manuscript and made some changes in the manuscript.  These changes will not influence the content and framework of the paper. And here we did not list the changes but marked in red in revised paper. We appreciate for Editors/Reviewers’ warm work earnestly, and hope that the correction will meet with approval. Once again, thank you very much for your comments and suggestions.

Best regards!

Yours sincerely,

Yan Yang E-mail:[email protected]

Name: Zhao-shi Xu E-mail: [email protected]
